# Short-term periodic restricted feeding elicits metabolome-microbiome signatures with sex dimorphic persistence in primate intervention

Hagai Yanai[1,6], Bongsoo Park[1,6], Hyunwook Koh [2], Hyo Jung Jang[2], Kelli L. Vaughan [1], Mayuri Tanaka-Yano[1], Miguel Aon [1], Madison Blanton[3], Ilhem Messaoudi [3], Alberto Diaz-Ruiz [4,5], Julie A. Mattison[1] & Isabel Beerman [1] ✉

Dietary restriction has shown benefits in physiological, metabolic, and molecular signatures associated with aging but is a difficult lifestyle to maintain for most individuals. In mice, a less restrictive diet that allows for cyclical periods of reduced calories mitigates aging phenotypes, yet the effects of such an intervention in a genetically heterogenous, higher-order mammal has not been examined. Here, using middle-aged rhesus macaques matched for age and sex, we show that a regimen of 4 days of low-calorie intake followed by 10 days of ad libitum feeding (4:10 diet) performed in repeating cycles over 12 weeks led to significant loss of weight and fat percentage, despite the free access to food for most of the study duration. We show the 4-day restriction period is sufficient to drive alterations to the serum metabolome characterized by substantial differences in lipid classes. These phenotypes were paralleled by changes in the gut microbiome of restricted monkeys that highlight the involvement of a microbiome-metabolome axis. This regimen shows promising phenotypes, with some sex-dimorphic responses, including residual memory of the diet. As many calorie restriction interventions are difficult to sustain, we propose that this short-term diet may be easier to adhere to and have benefits directly relevant to human aging.

Intervening against aging phenotypes by metabolic manipulation has been shown to be an effective strategy across a multitude of conditions and species[1,2]. The most common and well-studied intervention in this regard is continuous caloric restriction (CR), which emphasizes a daily reduction in the number of dietary calories consumed. Recent studies also suggest that in such dietary interventions, timing is an important dimension to consider, alongside caloric and dietary content, with prolonged periods of fasting contributing substantially to the beneficial effects of CR[3,4]. Indeed, some dietary interventions, including intermittent fasting (IF)[5] that capitalize on this, have been shown to be

[1]Translational Gerontology Branch, National Institute on Aging, NIH, Baltimore, MD, USA. [2]Department of Applied Mathematics & Statistics, The State University of New York, Korea (SUNY Korea), Incheon, South Korea. [3]Department of Microbiology, Immunology and Molecular Genetics, College of Medicine, University of Kentucky, Lexington, KY, USA. [4]Laboratory of Cellular and Molecular Gerontology, Precision Nutrition and Aging Program, Institute IMDEA Food (CEI UAM+CSIC), Madrid, Spain. [5]CIBER Physiopathology of Obesity and Nutrition (CIBERobn), Madrid, Spain. [6]These authors contributed equally: Hagai Yanai, Bongsoo Park. ✉e-mail: isabel.beerman@nih.gov

beneficial in ameliorating many pathophysiological indices in both rodents and humans[6,7]. Such interventions generally contain a segment of the day for fasting, and a period of regular ad libitum eating (AL), all occurring within a single circadian cycle[8], and these diets are often referred to as time-restricted feeding (TRF). Recent rodent studies indicate another iteration of this diet in which several days of reduced caloric intake followed by several days of AL could also be effective in regulating healthy metabolism, body weight, and healthspan[9]. In this design, the CR period occurs over several days, resulting in important differences compared to TRF in that the food restriction extends beyond a single circadian cycle[10], creating a prolonged period of altered gut absorption[11], and resulting in a prolonged metabolic shift[9]. We will refer to this type of diet regimen as periodic restricted feeding (PRF). As this strategy showed promise in rodents, we sought to test its translational potential in non-human primates (NHP), a model organism more closely related to humans. While CR and IF have shown promise in both humans[6,7,12] and NHPs[13,14], little has been described for the effect of PRF in either system.

Herein we explore the effects, efficacy, and safety of PRF in rhesus macaques (*Macaca mulatta*), a heterogenous NHP model with relevance to human physiology. We designed a short-term PRF regimen where caloric intake was restricted for 4 days, followed by 10 days of AL feeding (4:10 diet) for six cycles. Results suggest PRF is a viable, safe intervention for primates and provides insight into the effects of the diet on body composition, metabolic modulation, and associated gut microbiome alterations that are not solely driven by decreased caloric intake.

## Results

### Periodic restricted feeding results in loss of body mass and fat

Periodic restriction of caloric intake was evaluated in 12 male and 11 female adult rhesus monkeys divided into control (constant food access = adlib (AL)) and periodically restricted feeding (PRF) groups. To quantify baseline values, food consumption was monitored for 3 weeks prior to initiation of the study and animals were then assigned to either PRF or AL groups to best match for baseline age, sex, body weight, and fasting blood glucose levels to mitigate biases (Table 1). PRF animals were then calorically restricted for 4 days (50% restriction on day 1, 70% restriction on days 2–4), followed by 10 days of AL feeding (Fig. 1a). This process was repeated for 6 consecutive cycles, lasting a total of 3 months. No behavior abnormalities or outward signs of stress were observed by trained research technicians.

The most notable effect of PRF was a significant loss of body weight, both in absolute terms (delta from baseline; mean = 5% loss) and compared to AL controls (Fig. 1b). As all study animals remained in the animal colony post-dietary intervention, we were able to evaluate the weight differential between PRF and AL over 3 years after the study ended (Fig. 1b, right panel). We found that the loss of body weight was sustained over that period and this differential was independent of starting age (Suppl. Fig. 1a) or body weight (Suppl. Fig. 1b). A caveat to this long-term effect is that post-dietary restriction experiment, some animals were assigned to other studies. While these studies did not

include dietary interventions, there may be biases due to the study enrollments on the body weight measurements. We incorporated information regarding studies the PRF and AL monkeys were enrolled in: none were included in additional intervention studies and five (2 PRF, 3 AL) were included in an exercise study that may have affected body weight. We stratified the follow-up data into three categories: no additional study enrollment, non-intervention enrollment, and exercise study enrollment, and found the body weight differential remained regardless of study enrollment (Suppl. Fig. 1c).

A concern with PRF (or IF) studies is the potential for subjects to overeat (or gorge) during the AL period immediately following the restricted days. In this study, carefully measured food intake data from the PRF group during their AL phase of each cycle were statistically comparable to their baseline levels (and also similar to intake levels of the control AL group) suggesting that following the restriction period, animals did not overeat (Fig. 1c and Suppl. Fig. 1d). This effect was consistent across cycles (Fig. 1a), regardless of sex (Suppl. Fig. 1e) or age (Suppl. Fig. 1f). As a result, PRF animals consumed fewer total calories throughout the entire study compared to AL controls (Fig. 1d), with an overall effective CR of ~10-30% per animal (Suppl. Fig. 1g), contributing the overall weight loss. However, the effective CR measured for each animal was not significantly correlated with the change in body weight throughout the study (Fig. 1h).

In addition to body weight, to measure the overall impact of the diet, we quantified fat, bone, and lean-tissue body mass using dual X-ray absorptiometry (DEXA) scans (Suppl. Fig. 2a). PRF resulted in a loss of both lean and fat mass (Fig. 1e, f) with greater decreases seen in fat mass compared to lean mass (Fig. 1g). PRF-associated fat loss was ubiquitous to all body regions and did not alter fat distribution ratios (Suppl. Fig. 2b, c). Bone mineral density was also not affected by the PRF diet (Suppl. Fig. 2d). Together, the data show that this relatively short intervention of 4 days of CR followed by 10 days of unrestricted access to food, for six consecutive cycles, resulted in persistent loss of body weight without any observed harmful side effects.

### Cyclical PRF elicits a robust metabolic signature with the sex-dimorphic response after diet conclusion

To examine the metabolic impact of PRF, liquid chromatography–mass spectrometry (LC–MS) was used to successfully quantify 866 metabolites from blood serum at the indicated time points (Fig. 2a). Principal component analysis (PCA) indicates the global effect of the diet on metabolism is predominantly transient (Fig. 2b and Suppl. Fig. 3a) with a shift in PC1 occurring at peak restriction (day 4) in both cycle 3 and cycle 6 but returning to near baseline after refeeding (day 14).

To better understand the core metabolic signature of the cyclical PRF intervention, we examined differentially abundant metabolites ($p < 0.05$, $q < 0.1$, FC > 30%) at peak restriction in cycles three and six (C3d4, C6d4: Fig. 2c) and found a significant number of shared modulated metabolites between the cycles (45 of 102 up, and 11 of 78 down) that were common to both males and females, representing the core metabolic response (Fig. 2d). The up-modulated metabolites were classified mostly as different lipid classes (Fig. 2c). Among them, the top-ranked enriched metabolites were the fatty acids acyl-, esters- and conjugate forms, followed by sphingomyelins, phospho-sphingolipids, glycerophospholipids and glycerol phosphocholines (Fig. 2e, f). Importantly, the presence of the ketone body 3-hydroxybutyrate and medium and long chain fatty acids, as well as several carnitine conjugates (Suppl. Data 1) suggests these compounds serve as energy sources during PRF, as carnitine derivatives are readily available due to their ability to permeate membranes. Another class of increased lipids in the PRF group were those conjugated with glycine (Suppl. Data 1) such as 2-butenoylglycine, 3-hydroxybutyroylglycine,

**Table 1 | Demographic information of the study participant**

|  | AL | PRF | Both |
|---|---|---|---|
| Sex (M/F) | 6/5 | 6/6 | 12/11 |
| Age (years) | 14.37 | 13.52 | 13.93 |
|  | (± 1.54) | (± 1.72) | (± 1.14) |
| Weight (kgs) | 10.13 | 11.27 | 10.72 |
|  | (± 0.61) | (± 0.73) | (± 0.48) |
| Fasting Blood Glucose | 67.82 | 67.50 | 67.65 |
|  | (± 2.08) | (± 2.05) | (± 1.43) |

The values represent the mean (± SEM).

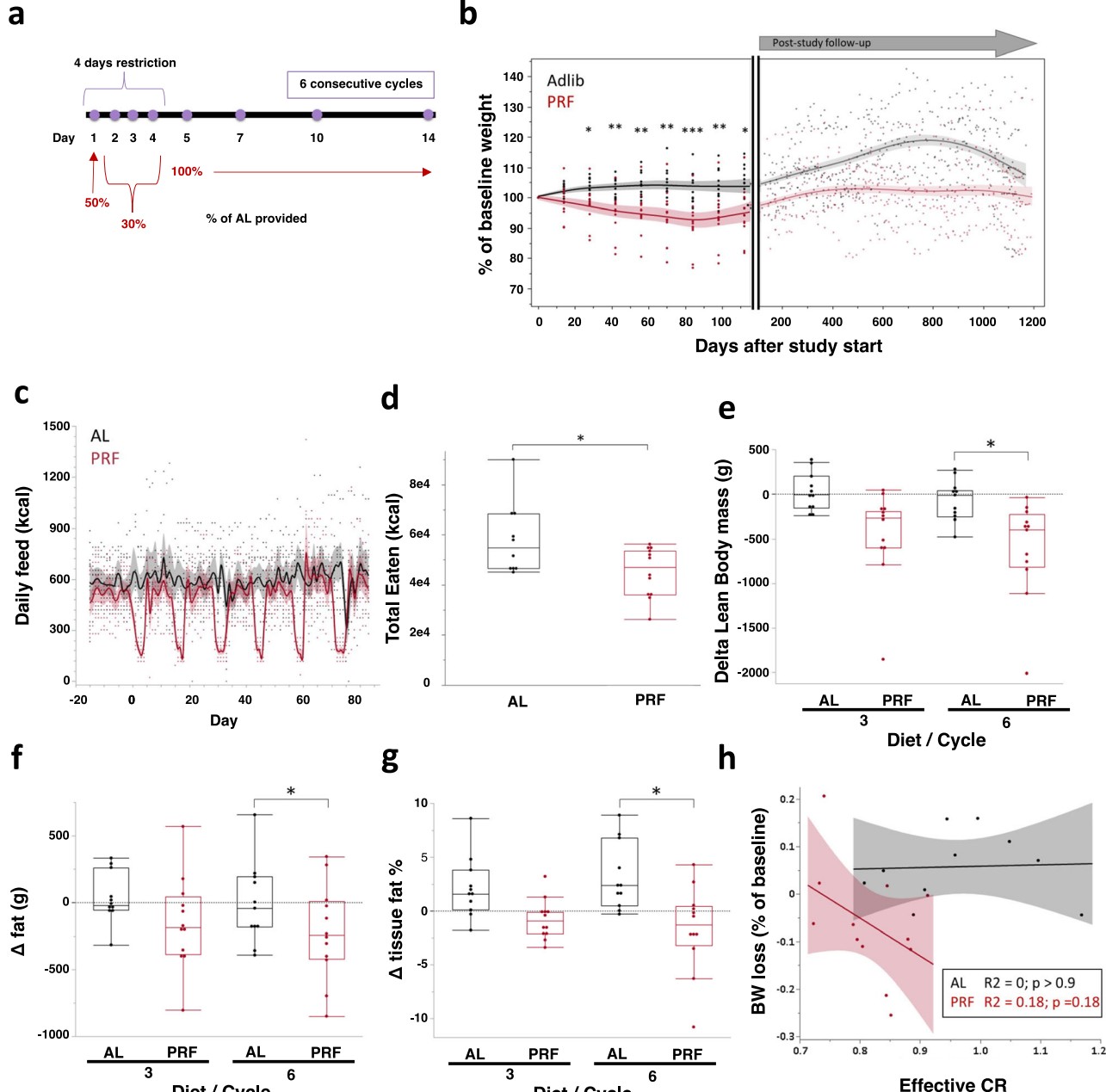

**Fig. 1 | Impact of PRF on feeding, body weight, and body composition. a** Diet design (generated with BioRender). **b** Body weight presented as a percent of baseline body weight. Each dot represents a single measurement, the group line was calculated with a spline function ($\lambda = 0.5$) and the shaded area represents the confidence of fit with a color corresponding to the experimental group. Statistical difference was performed by two-way ANOVA coupled with Tukey's post-hoc test with *p*-values as described below. **c** Food uptake. Dots and lines are as in (b). **d** Total calories eaten by each animal throughout the entire study period. $p = 0.0257$ in a two-tailed *t*-test. **e**, **f** Changes measured by DEXA scan, presented for each animal as a delta from baseline to the indicated time points for lean body mass ($p = 0.0074$) (**e**), fat mass ($p = 0.0005$) (**f**), fat % ($p = 0.0037$) (**g**). Data are represented as box and whisker plots, depicting minimum, lower quartile (Q1), median (Q2), upper quartile (Q3), and maximum values. Two-way ANOVA coupled with Tukey's post-hoc test [diet (AL, PRF), cycle number (3 vs. 6), and their interaction]. *, $p < 0.05$; **, $p < 0.01$; ***, $p < 0.001$. **h** Effective CR was calculated for each animal as average daily feed throughout the study divided by the baseline daily feed. Plotted is a linear correlation plot of effective CR vs. the body weight change throughout the study. Each dot represents a single animal with a linear regression (line) and confidence of fit (shaded area). Statistics for the linear correlation are denoted on the plot. For the entire figure $n = 12$ for PRF and $n = 11$ for AL. For detailed data points please see Source Data file.

hexanoylglycine, and N-octanoylglycine. We noticed a marked alteration of structural membrane components that suggests altered membrane remodeling processes. This includes an increase in glycerol-phospholipids and -phosphocholines along with sphingomyelins, and correspondingly 10 of the 11 decreased metabolites were glycerophospholipid membrane components (Fig. 2f). While a significant number of metabolites were uniquely altered during cycle 3

(Fig. 2c, d), these metabolites belong to lipid classes that are similar to the shared metabolites defined above, suggesting a consistent and concerted metabolic response (Suppl. Fig. 3g–i).

A murine intervention with similar cyclical restriction in middle-aged mice reported persistent metabolic changes following PRF[9], so we also analyzed the metabolome profile for a potential metabolic memory of the NHP group. At the end of the study, the metabolic

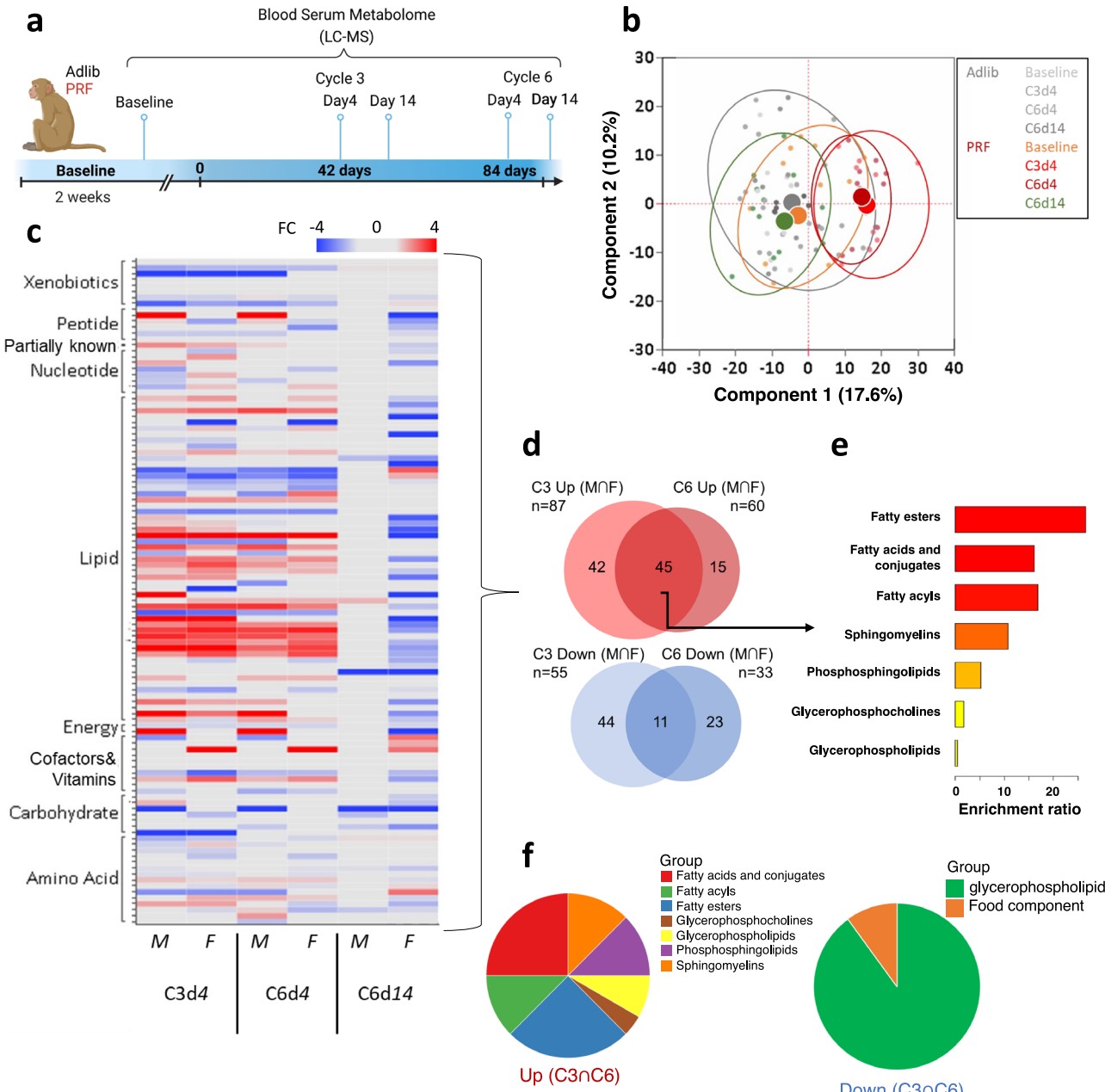

**Fig. 2 | Metabolomic profile. a** Blood serum collection timepoints for metabolomic profiling (generated with BioRender). **b** Principal component analysis (PCA) using all 866 metabolites measured. Each dot represents a single animal at the indicated time point. Ellipses contain all individuals of the indicated group, and the large dots represent the centroid for each group, with AL shown as a single group. **c** Heatmap of metabolites with differential abundance during PRF vs. AL compared to baseline. ($n = 12$ for PRF and $n = 11$ for AL). Only metabolites with significant changes in response to PRF at day 4 during cycles 3 and 6 ($p < 0.05$, $q < 0.1$, FC > | 1.3 | ) are presented. M, males ($n = 12$); F, females ($n = 11$). For detailed statistics please see Source Data file. **d** Venn diagram depicting the impact of sex (M vs. F) on metabolite abundance in the PRF vs. AL pairwise comparisons at day 4 during cycles 3 (C3) and 6 (C6). **e** Enrichment analysis of shared metabolites upregulated in PRF-fed males at day 4 during C3 and C6. **f** Pie charts depicting the distribution of the shared metabolites by type that were up- (45) and downregulated (11), during PRF.

signature driven by the PRF diet was reset to baseline in males but over-corrected in females (Fig. 2c). This sex-dimorphic over-correction of the female metabolome signature consisted of 31 increased and 134 decreased metabolites with a strong inverse overlap of the signature observed at day 4 of the diet (Fig. 2c and Suppl. Fig. 3f). For example, sphingolipids such as sphinganine, sphinganine-1-phosphate, and sphingadienine were all strongly downregulated at the end of the diet (C6d8), whereas they were all robustly increased at peak diet (C3d4, C6d4) compared to baseline. In addition, we observed a reduction of amino acids such as glycine circulating in the serum (Suppl. Data 1), which were not increased during peak diet.

Together, the results show that a serum metabolome consisting of 56 up- and down-modulated metabolites from mostly different lipid classes represents a metabolic signature of PRF. Unlike males, an inverse female metabolic signature emerged after the conclusion of the dietary intervention unveiling a dimorphic response. This sex-specific response was consistent with the long-term body weight follow-up in which the inverse metabolite signatures in females at the end of the diet were associated with regaining weight lost during cyclical PRF rather quickly, whereas the males, which did not over-correct their metabolome, sustained their PRF-associated weight loss long-term (Suppl. Fig. 4).

## The gut microbiome rapidly responds to PRF in both the restricted and AL cycles

During uninterrupted CR, the gut microbiome adapts to altered nutrient amounts[15–17]. However, it is unclear how the microbiome reacts to a cyclical CR regimen. We examined the gut microbiome using fecal 16S RNA sequencing at the indicated time points to account for changes during the restricted and AL eating phase (Fig. 3a). Analyses of multiple α-diversity indices, which quantify the abundance and frequency of the operational taxonomic units, show trends of increased α-diversity during peak diet after three PRF cycles (C3d4), compared to AL microbiomes at the same time point (Fig. 3b). By cycle 6, this trend becomes statistically significant for most indices, suggesting an escalation of the response with cumulative cycles (Fig. 3b). This trend is also apparent when PRF and control AL microbiomes are compared to their baseline (rather than to each other) with PRF C3d4 and C6d4 showing increased alpha-diversity during peak diet restriction and maintaining an increased diversity after refeeding (C6d8) (Fig. 3c). In contrast, AL animals did not show any increases in α-diversity throughout the experiment.

While α-diversity measures diversity within groups, β-diversity examines the similarity or lack thereof between groups. β-diversity analysis, visualized by principal coordinate analysis (PCoA), shows inter-individual heterogeneity of the rhesus monkey gut microbiome dominates the analysis, resulting in only weak statistical significance of β-diversity between the PRF and AL samples (Fig. 3d). Despite the heterogeneity, a differential abundance analysis reveals statistically significant changes in specific taxonomic units during peak PRF diet (day 4) (Fig. 3e). Importantly, these altered taxonomies were concordant between cycles 3 and 6, suggesting a systematic response to the caloric restriction. The most prominent and consistent alterations observed were the increase in Verrucomicrobia and the closely related Lentisphaerae phyla alongside a decrease in Firmicutes.

The consistent response to PRF is also evident in a PCoA analysis of the microbiome in only PRF animals, where microbiome diversities cluster together based on cycle phase (day) rather than cycle number (Fig. 3f), suggesting the cyclical nature of the diet is associated with a rapid shift in the gut microbiome profile. This effect is exemplified in the abundance of the Verrucomicrobia phylum which remains stable in AL control animals but increases during the CR phase of the PRF diet, returning to normal levels during the AL refeeding period (Fig. 3g). In addition, the mean level of abundance of the Verrucomicrobia was higher on cycle 6 compared to the cycle 3, suggesting repeated cycles of CR create a cumulative response to the same challenge.

To examine if any microbiome alterations were maintained after the last cycle of PRF, we examined the differential abundance of OTUs in the microbiome of PRF animals at the end of the last cycle (C6d8) compared to baseline. Indeed, PRF subjects maintained an elevated abundance of Verrucomicrobia and Lentisphaerae even after returning to AL consumption (Fig. 3h).

Previous reports have demonstrated key interactions between the gut and immune systems; however, in this study though we had significant changes in the microbiome, we did not observe significant alterations among the 36 blood parameters we measured using flow cytometry immunophenotypes and complete blood count data compared to baseline (Suppl. Figs. 6–8). However, when looking for impact of the diet using unsupervised single cell analysis, Citrus[18], we did observe an increase in a subpopulation of Neutrophils alongside a decrease in a subpopulation of CD4 T-cells (Suppl. Fig. 9). But overall, a principal component analysis (PCA) and multivariate analysis of the observed changes indicates that throughout the study, PRF animals presented a more stable blood profile compared to those in the AL control group (Suppl. Fig. 9). To evaluate if PRF affected the inflammatory status, we screened 36 cytokines in the serum and determined the PRF diet had no significant impact on the inflammatory status of the NHP (Suppl. Fig. 10).

Finally, we sought to predict a functional profile of the PRF microbiome by evaluating changes to metabolic function using PICRUSt2[19]. We found an abundance of pathways that are uniquely enriched during the PRF peak diet (day 4) in both cycles 3 and 6 (Fig. 4a and Suppl. Data 2). Interestingly, many of these metabolic pathways overlap with the metabolome fingerprint of the PRF intervention (Fig. 2), including those related to lipid metabolism (e.g., coenzyme A and pantothenate, beta-oxidation, fatty acid elongation, phospholipids, and diacylglycerol), aerobic and anaerobic bacterial metabolism (e.g., pyruvate and lactate fermentation, flavins, and folic acid), and pathways generating intermediates of the folate cycle. When testing for metabolic pathways enriched after the refeeding of the final cycle (C6d8), we found a sex-dimorphic phenotype, where female PRF animals do not any enrichments not also seen in AL subjects, whereas the males present several uniquely enriched metabolic pathways (Fig. 4b) including a decrease of TCA cycle VII (PWy-7254), and an increase in synthesis of ADP-L-glycero-beta-D-manno-heptose (PWy-1241) and 4-aminobutanoate degradation to butyrate (PWy-5022).

Taken together, these results suggest that limiting calorie consumption in a cyclical manner, without altering the composition, leads to swift changes in the microbiome that appear to be additive and persist after the refeeding in the last cycle. These changes correspond to the alterations observed in the serum metabolome and suggest a gut microbiome-metabolome axis is modified in the short dietary intervention associated with overall weight loss.

## Limitations of study

This study contains limitations which we attempted to account for when interpreting the results. In order to control feeding, animals in the study were single housed, but all study subjects were subjected to the same housing and thus housing should not have adversely affected the comparisons. While we reported on the immune profile using leukocyte counts, frequencies, and circulating cytokine levels, we could not perform functional immune response tests due to COVID restrictions at the time. Considering the natural heterogeneity in NHP colonies, ideally, we would have included a larger number of animals in the study as well as including a more diverse range of ages, to include young and aged NHPs. However, due to the availability of animals and challenges with the coordination of space and feeding schedules for all subjects, we have only included middle-aged monkeys in this study. Finally, the observation made on the preservation of body weight loss was made retrospectively and some NHPs were enrolled in other studies. However, the distribution of the animals enrolled in studies that may have altered BW was similar even (3 AL and 2 PRF monkeys) and regardless of post-PRF grouping (no study, non-intervention study, or potential intervention study), the PRF-male BW loss was maintained.

## Discussion

In this study, adult rhesus monkeys underwent six consecutive cycles of PRF. Cycles included 4 days of severely reduced caloric intake followed by 10 days of unrestricted feeding (AL). Despite the caloric reduction during the 4-day restrictive phase, PRF animals did not overindulge in calories (i.e., overeat) during the AL phase to compensate for the deficit. This behavior resulted in an overall reduction in caloric consumption over the course of the experiment and a significant weight loss compared to AL controls. However, one interesting observation from this study is the lack of direct correlation between the effective CR (the actual amount of reduced food intake) and the loss of body weight, suggesting the loss of body weight with this PRF regimen is a different paradigm than simply the direct effect of reduced calories. It would be interesting to see if the missing element is the individual energy expenditure and whether that expenditure difference is due to basal metabolic rate or activity differences.

Further, during the 4-day CR phase of the cycle, the metabolomic profile suggests PRF animals strongly activated pathways of lipid

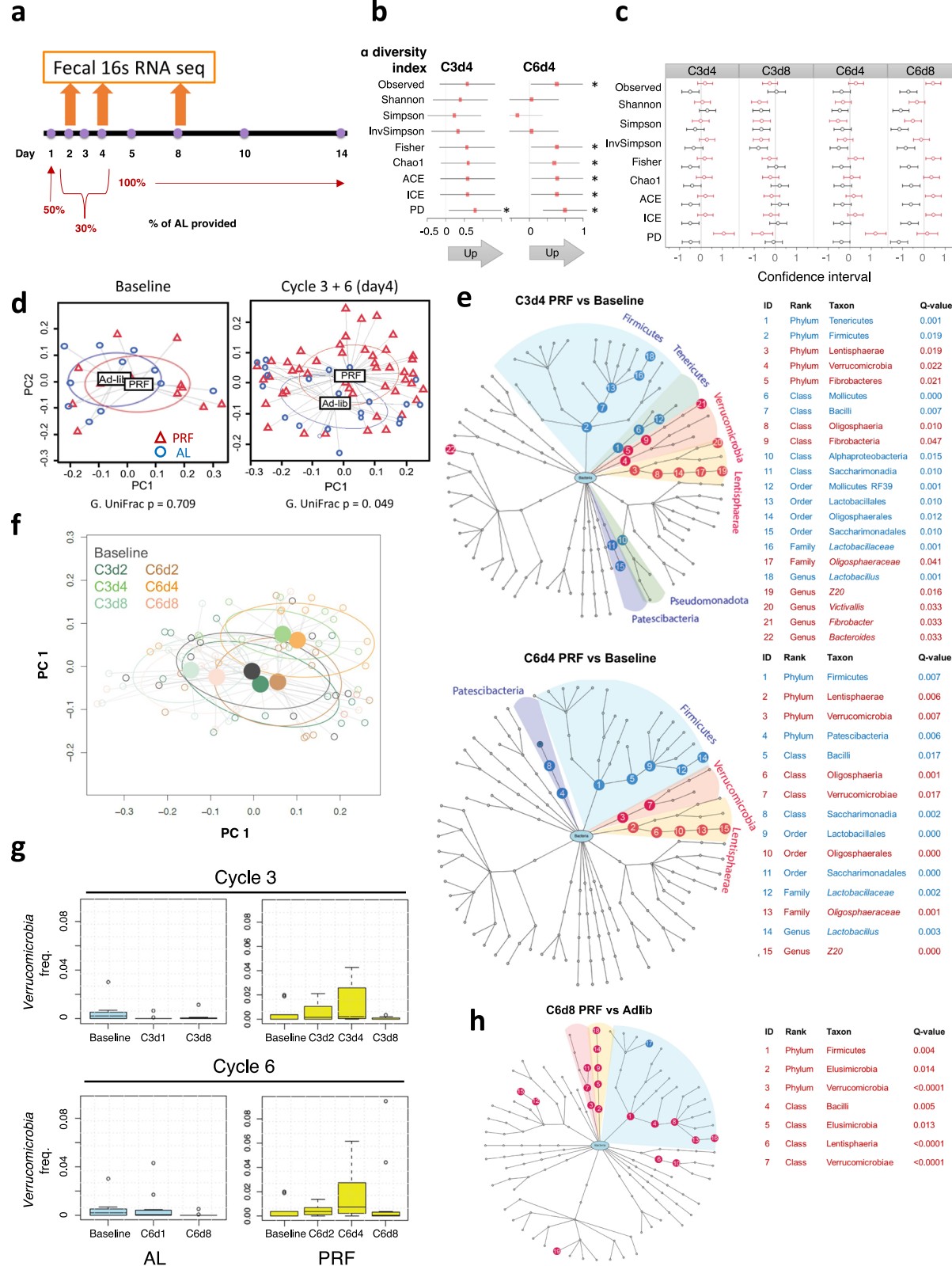

**Fig. 3 | Microbiome changes as a result of the PRF diet. a** Timepoints of fecal collection for 16S RNA sequencing. **b** α-diversity comparison between PRF (*n* = 12) and AL (*n* = 11) at 4 day of cycle 3 (C3d4) and cycle 6 (C6d4). **c** α-diversity comparison of PRF (*n* = 12) and AL (*n* = 11) to baseline at the indicated time points. Black and red lines represent the AL and PRF groups, respectively. **d** Principal coordinates analysis (PCoA) plots for the AL (blue circles, *n* = 12) and PRF (red triangles, *n* = 12) groups at baseline, C3d4, and C6d4. The ellipses correspond to 95% confidence intervals for a normal distribution. The *p*-value represents the generalized UniFrac beta-diversity comparison score. **e** Differential abundance analysis of PRF animals (*n* = 12) at C3d4 (upper panel) and C6d4 (lower panel) vs. baseline, presented with a hierarchical tree. Specific increased (red) and decreased (blue) OTUs are listed in the right panel. **f** PCoA of PRF animals at baseline and at the indicated timepoints. The ellipses correspond to 95% confidence intervals for a normal distribution and centroids for each experimental group are depicted. **g** Abundance of Verrucomicrobia under AL and PRF regimen. **h** Differential abundance analysis between PRF (*n* = 12) and AL (*n* = 11) animals at the C6d8 time point.

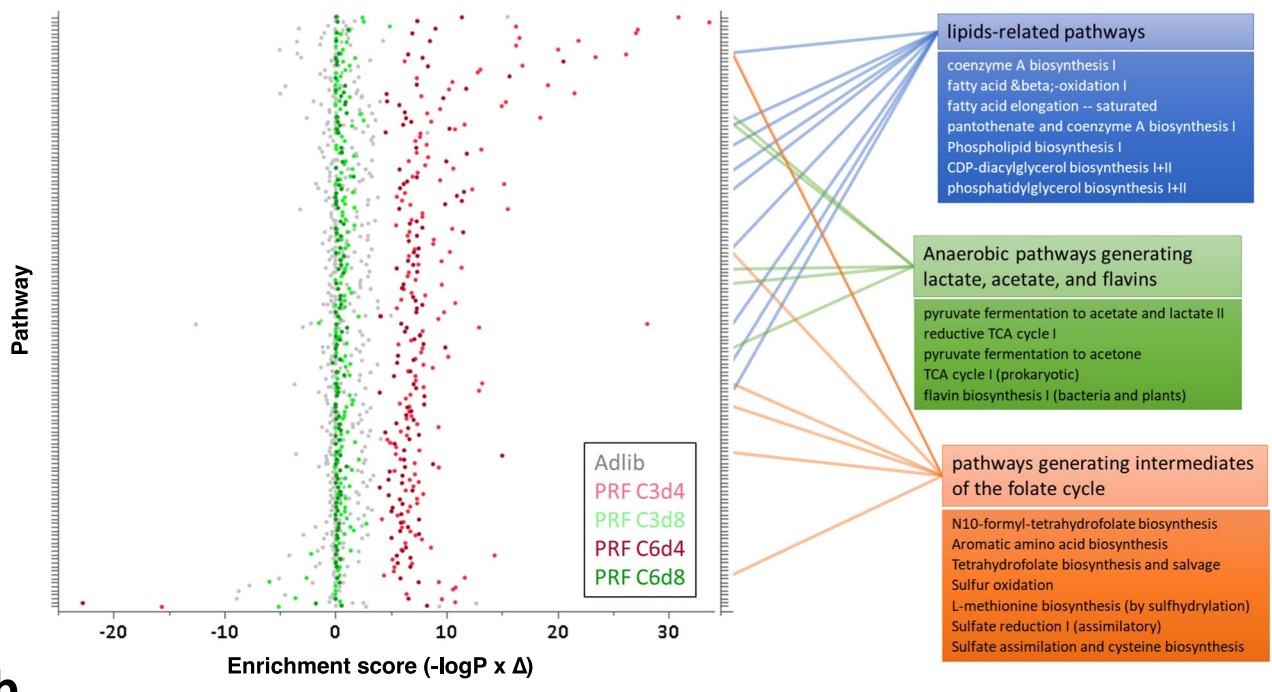

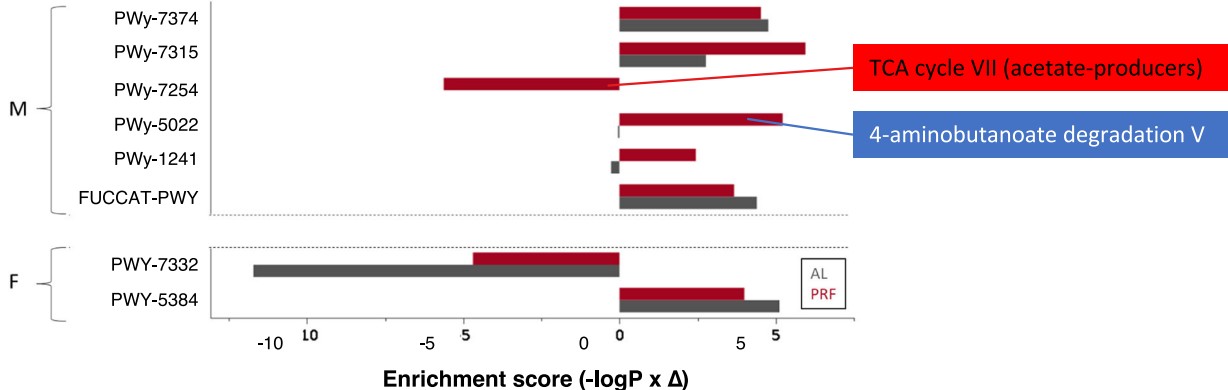

**Fig. 4 | Enriched metabolic pathways within the gut microbiome profile.**
**a** PICRUSt2 analysis showing enrichment of metabolic pathways at C3d4 and C6d4. The dot plot summarizes all enriched pathways ($p < 0.05$) as a function of group (denoted by color, $n = 12$ per group) and the combined enrichment score (calculated as $-\log (p)$ x delta). The right panel highlights pathways relevant to the observed metabolome changes (Fig. 2). For statistical details please see Supplementary Data 2. **b** PICRUSt2 analysis for C6d8 separated for sex and presented as a horizontal bar graph separated for AL (dark gray) and PRF (red).

utilization, also resulting in a loss of fat percentage that was independent of body region. The loss of body weight was also associated with loss of lean mass but without any observed or quantified negative side effects. These results are highly consistent with outcomes reported from a study administering an identical diet regimen to middle-aged male C57Bl6/J mice[9].

We found that PRF subjects, especially males, maintained a significantly lower body weight compared to the AL controls for many months following the study's conclusion. This is observational, as some subjects were subsequently assigned to other projects and not internally controlled for this question. Upon stratification of the studies the monkeys were enrolled in, we still observed this maintained

body weight loss in the males. This weight loss was present even though in males the metabolic profile at the study end (as measured in the serum) was restored to levels close to the baseline. In contrast, in females subjects the divergence of body weight between the PRF and AL monkeys was rapidly lost, and there was an overcompensation of the metabolic profile shifts at the study end—i.e., profiles upregulated compared to baseline during the dietary restriction became down-regulated compared to baseline, rather than returning to baseline levels. Thus, we speculate the weight gain in the females may be correlated to the metabolome over-correction, but we cannot state that for certain as we did not measure body composition after the study ended to determine if the regaining of the weight in females was due to

fat and/or lean mass accumulation associated with the metabolism shifts. Regardless, the sex-dimorphism in the lasting response to the diet suggests females present a more reactive metabolic system with metabolite changes overcompensating in response to the PRF after the females resume a regular AL diet, whereas the PRF modifications of the metabolome in males returned to baseline levels. The lack of a significant "metabolic memory" of the diet in male monkeys, which was observed in the mouse study using the same dietary restriction[9], may be attributed to the duration of the intervention compared to the subjects' lifespan. The mouse study lasted 5 months (compared to a ~3-year lifespan) compared to only a 3-month intervention in the macaques (compared to a ~30-year lifespan). This duration discrepancy could partially explain the stronger metabolic memory observed in mice compared to the monkeys. Another hypothesis that could explain the observed retention of body weight loss could relate to a potential impact of the microbiome on epigenetic regulation in either intestinal cells or other peripheral tissues[20]. This would be especially interesting in the context of influencing the endocrinological system in a sex-specific manner but that remains to be studied.

Of note, like humans, NHPs are inherently diverse, leading to high levels of heterogeneity in both the baseline and the observed response outcomes for virtually all parameters. Importantly, even with this heterogeneity and relatively small sample size, we were still able to discern phenotypes, some surpassing significance thresholds, supporting the robustness of the dietary intervention. Towards resolving the nuanced mechanisms underlying these alterations, we provide initial data to help establish power calculations for the number of subjects required for future studies, especially if sex-specific parameters are to be further evaluated.

Considering that the profile of circulating blood cells is sensitive to physiological conditions and these population frequencies change during sustained CR[21,22] and aging[23] in multiple organisms, we expected to observe changes in response to PRF but could only detect changes in subpopulations of $CD16^{high}$ Neutrophils and CD4 T-cells with an unsupervised analysis. We found the blood signature remained surprisingly stable in the restricted group, even mitigating the increased heterogeneity observed in the AL control group throughout the duration of the study. This reduction in heterogeneity was not associated with an altered profile of circulating cytokine levels. The homeostasis sustained in the PRF group may be attributed to the cyclical nature of the diet, allowing for periods of AL refeeding in contrast to studies of continuous CR, allowing for higher retention of immune function in dietary-restricted animals. It is important to note, however, that we were not able to perform immune challenge studies or functional tests on these subjects following the study, which would be required to better define the impact of PRF on the immune system.

Surprisingly to us, the animals in this study presented with high inter-individual variability of the gut microbiome even though the housing environment and diet were identical, which could have made the impact of the diet difficult to ascertain[24]. Despite this heterogeneity, we observed the microbiome profile changed in a robust, consistent, and rapid (2–4 days) fashion in response to the decrease in calories consumed. The altered α-diversity indexes suggest that the impact of PRF is generally positive, as increased diversity is generally associated with better health[25]. The interesting phyla that consistently changed under PRF were the Verrucomicrobia and Firmicutes. In humans, obesity has been associated with low levels of Verrucomicrobia[26], which increased as a result of PRF in rhesus macaque. Transplantation with the Verrucomicrobia *Akkermansia muciniphila* in progeroid mouse models is sufficient to enhance healthspan and lifespan[27]. Our study shows an augmentation of Verrucomicrobia levels in rhesus macaque as a result of PRF, thus constituting a PRF-mediated mechanism to promote overall health. Conversely, PRF promoted a reduction of Firmicutes, which are reported to be increased in obesity and type 2 diabetes[28]. While the

microbiome changed rapidly in response to diet in the PRF group, the profile also rapidly returned to almost normal levels after refeeding. However, we did observe a slightly altered profile at the conclusion of the study suggesting closer examination using a larger number of subjects, extended number of consecutive PRF cycles, and deeper sequencing could potentially confirm the existence of microbiome memory. Another potential factor that could play an important role is the impact of PRF on the circadian rhythm and the gut microbiome[29,30]. While we did not closely monitor sleep patterns or measure circadian gene expression and therefore cannot make definite conclusions, we posit that the circadian rhythm was minimally perturbed.

At present, we cannot exclude the possibility that microbiome components are a source of at least some of the metabolites present in the serum; for example, pathways from bacterial metabolism generating metabolites like lactate and acetate or short-chain fatty acids like butyric and propionic or, eventually, flavins (Fig. 4a). Accordingly, metabolism of propionate, butyrate and essential polyunsaturated fatty acids has been implicated in caloric restriction and is known to influence immunity and inflammation in mice and humans[12,31]. Importantly, the systemic beneficial effects of propionic acid have been linked to the gut microbiome via the degradation of dietary fiber intake, a primary substrate for propionic acid generation[32]. In addition, we find preliminary evidence that the microbiome profile in the males could, at least in part, mediate retaining the loss of body weight for a long period after the intervention ended. Considering that the microbiome has been previously shown in humans to regulate gut absorption[33], it is tantalizing to consider this as a potential mechanism to explain both the retention of BW loss and the lack of correlation to effective individual CR.

In summary, we found that short-term, consecutive PRF cycles result in a significant loss of body weight and fat percentage in adult rhesus monkeys. This was accompanied by complimentary changes to the gut microbiome and the metabolic profile, with stable hematopoiesis and without discernable negative side effects. Future studies involving altered composition of the diet, varying levels of restriction, and increased numbers of cycles will help to further characterize the benefits conferred by PRF. We propose these data are promising for translation of this type of intervention for humans, as one major challenge to strict dietary interventions in the heterogenous human population is the lack of adherence to sustained, severe caloric restriction: the short, cyclical nature of this intervention may be easier for humans to follow, while still providing significant benefits.

## Methods

We confirm that the presented research complies with all relevant ethical regulations as instructed by the National of Health Intramural program regulations. All experimental protocols were approved under protocol # 434-TGB-2024.

### Animals

Animals were maintained at the NIH Animal Center and housed in standard primate caging with controlled temperature and humidity and 12-h light/dark cycles. All animals received a commercially prepared monkey chow (Purina Mills, St. Louis, MO) at approximately *ad libitum* levels twice per day (at 07:00 and 14:00) along with a daily food enrichment item provided on a rotating schedule (identical between AL and PRF). Water was available *ad libitum*. All procedures were performed in accordance with the Guide and approved by the National Institute on Aging's intramural animal care and use committee. Subjects were 12 male and 11 female rhesus monkeys aged 7 to 14 years ($M = 13.93 \pm 1.14$) (Table 1). Groups were matched based on sex-, age-, body weight-, and fasting blood glucose assessed at baseline. To manage labor and experimentation scheduling, animals were divided into 4 overlapping cohorts tested on a staggered schedule. For a full

schedule Gant please refer to Suppl. Fig. 11. All experimental protocols were approved under protocol # 434-TGB-2024.

## Diet

Food intake was monitored individually for three weeks prior to the start of the study; AL food volumes were established during this time. Monkeys were fed two meals per day at 6:00 AM and 1:00 PM and at each feeding, technicians confirmed that all animals had at least one biscuit remaining from the previous meal. The AL control group animals remained on the twice-per-day feeding schedule throughout the study. All animals received a standard commercially prepared chow: LabDiet 5038, 5045, or 5049 (LabDiet®, St. Louis, MO), all of which complete all nutrient requirements.

Following baseline procedures, animals in the PRF group underwent six consecutive 14-day cycles (Fig. 1). Specifically, on day 1 of each cycle, each animal received 50% of its baseline established AL volume of food provided as a single meal administered mid-day. On days 2–4, the volume of daily feed was further reduced to 30% of the baseline AL caloric intake, also provided once per day. On days 5–14, PRF animals were transitioned back to their AL volume plus 50%. To reduce binge feeding on days 5 and 6, rations were evenly divided and offered in small allotments provided approximately every 2–3 h throughout the day. The length of time given for each small meal, the number of biscuits fed, and the number of meals modified in this manner were based on each animals' individual behavior, as determined by trained technicians. In general, by day 7, feeding behavior had returned to baseline values and two meals per day allotments resumed. Following 10 days of AL feeding, the PRF cycle was repeated for a total of six cycles lasting a total of 12 weeks. Throughout the study, controlled enrichment was provided daily and matched with controls. However, on PRF days, enrichment food volumes were likewise reduced. Food consumption for each animal was monitored continuously throughout the study and recorded for three weeks at baseline and then twice daily for the remainder of the study.

## Blood collection and analysis

Blood samples were obtained under ketamine (7–10 mg/kg, IM) or telazol (3–6 mg/kg, IM) following an overnight fast. Procedures were performed at baseline and repeated following cycles three and six of PRF. Serum and plasma were separated by centrifugation, frozen, and stored at −80 °C until analyzed. Serum and whole blood were sent to Antech Diagnostics (Irvine, CA, USA) to assess blood chemistry and complete blood count values and to Metabolon (Durham, NC, USA) for LC−MS metabolome profiling. Hemoglobin A1C assays were performed using the Siemens DVA Vantage® analyzer to obtain a quantitative measure of the percent concentration of HbA1C in blood. All physiological measurements were collected within 1 h of sedation.

A 4 mL sample of whole blood was kept in an EDTA tube on ice for flow cytometry. Briefly, blood was treated with ACK to remove RBCs, and stained with conjugated monoclonal Abs (CD20-APC.Cy7, Biolegend clone 2H7; CD3-PE.Cy7, BD clone sp34-2; HLA-DR-ECD, BD clone immu-357; CD14-BV421, BD clone M5E2; CD8-APC, Biolegend clone SK1; CD4-FITC, BD clone L200; CD16-PE.Cy5.5, Biolegend clone 3G8; CD1c-PE, Militenyi Biotech clone AD5-8E7; CD123-PerCP.Cy5.5, BD clone 9F5) and Aqua-live/dead-BV510 (Thermofisher Scientific, L34597) for 30 min on ice, and measured on a BD FACSAria™ Fusion. Analysis was performed with FlowJo (BD, NJ, USA).

## Serum metabolomics and cytokine analysis

All serum metabolomics were performed by Metabolon (Durham, NC, USA), measuring 874 metabolites by LC−MS. Metabolites were then batch normalized (divided by the median of each batch) and batch imputed (lowest value). For differential analysis, data was transformed by natural log and assessed by ANOVA, corrected for multiple comparisons (FDR).

All serum cytokine levels were quantified using the Non-Human Primate XL Premixed Luminex Performance Assay (RnDSystems, Cat# FCSTM21-36) according to the manufacturer's instructions.

## Fecal collection and microbiome analysis

Fecal samples were freshly collected on the indicated time points from the animal's home cage and frozen in 5 mL cryovials for subsequent microbiome analysis. 16S V3-4 Amplicon-Seq with 30%PhiX was performed on a MiSeq instrument by the JHU Single Cell & Transcriptomics Core.

Raw sequencing data was processed using the QIIME2 pipeline, and the SILVA microbiome database (16S rRNA RefSeq version 3.2.1). DADA2 and USEARCH were used to remove sequencing errors and chimeras. The microbiome data originally included 651 amplicon sequence variants (ASVs) for 139 samples (55 AL for 11 monkeys, and 84 PRF samples for 12 monkeys), but we applied quality controls removing subjects with a total read count <2000 and ASVs that have a mean proportion <0.00002; as such 447 ASVs for 139 samples were finally retained in the analysis.

Nine α-diversity indices were calculated (i.e., Observed, Shannon, Simpson, Inverse Simpson, Fisher, Chao1, ACE, ICE, PD) using the R packages aMiAD, fossil, picante, and entropart. Paired baseline and time point microbiome analysis was performed by parametric paired $t$-test. For cycle 3 and 6 time-point analysis, we also fitted the random effects model to assess the disparity in each α-diversity index between AL and PRF-treated NHPs while adjusting for age, sex, and blood glucose. We used the Wilcoxon signed-rank test to estimate the disparity in each α-diversity index between AL and PRF-treated microbiome samples.

Five ecological distance metrics were calculated (i.e., Jaccard dissimilarity, Bray-Curtis dissimilarity, Unweighted UniFrac distance, Generalized UniFrac distance ($\theta = 0.5$), and Weighted UniFrac distance) using the R packages, GUniFrac and MiRKAT. We used GLMM-MiRKAT to estimate the disparity in each distance metric between AL and PRF-treated microbiome samples.

To calculate the differential abundance of taxonomic units, we applied the centered log-ratio (CLR) transformation to relax the compositional constraint. Taxonomic differential abundance analysis was done between the baseline and each of the other time points with respect to each taxon at different taxonomic ranks (i.e., phylum, class, order, family, genus) for the AL and PRF groups, respectively. We applied the Benjamini−Hochberg (BH) procedure per taxonomic rank to set a false discovery rate (FDR) threshold of under 5%. Dendrograms representing discovery status from the taxonomic differential abundance analysis were generated with R.

Functional analysis of microbiome metabolism was performed using PICRUSt2 to generate functional annotations (i.e., KEGG pathways). Then, we applied the CLR transformation using Python. For paired baseline and time point microbiome analyses, we used a parametric paired $t$-test and applied the BH procedure per KEGG pathway rank to set a control FDR threshold of under 5%.

## Statistics and reproducibility

Sample sizes in this study were determined according to animal availability rather than predetermination by statistical methods and included 12 male and 11 female rhesus monkeys aged 7 to 14 years ($M = 13.93 \pm 1.14$) (Table 1). Groups were matched based on sex-, age-, body weight-, and fasting blood glucose assessed at baseline. Initially, 12 females were included, however, a single female was excluded from the study due to medical complications. All data analysis was blinded. All statistical analyses in the manuscript are described in the corresponding figure or Method section. All other analyses have been performed on the JMP: Statistical Software platform (JMP, NC, USA).

**Reporting summary**

Further information on research design is available in the Nature Portfolio Reporting Summary linked to this article.

## Data availability

The microbiome data generated in this study have been deposited in the GEO database under accession number GSE235769. The data is freely available according to NIH guidelines. The metabolomic data generated in this study is available as Supplementary Data 1. Pathways enriched in the PRF subjects' microbiome can be found in Supplementary Data 2. Data for figures is included in the Source Data file. Any other data not presented in the article will be readily provided by request. Source data are provided in this paper.

## Code availability

All codes used in the study are available on GitHub https://github.com/yj7599/mipairgit.

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

## Acknowledgements

We would like to thank Drs Rafael de Cabo, Michel Bernier, and Nate Price for their thoughtful discussions and editorial comments. We would also like to thank and acknowledge the team at the NIA NHP Core, especially Kielee Toepfer, Caryn Seward, Ed Tilmont, and Rick Herbert DVM. This work was supported by the Intramural Research Program of the National Institute on Aging, a grant of the Comunidad de Madrid-Talento Grant (2018-T1/BMD-11966), Spanish Agency of Investigation (AEI /10.13039/501100011033), Ramon y Cajal Award from the Spanish Ministry of Science, Innovation and Universities (MICINN) (RYC2021-033751-I), RETOS Projects Program of MICINN (PID2019-106893RA-I00), and Ramón Areces Foundation (CIVP21S13338) (A.D.R). H.K. is supported by the National Research Foundation of Korea (NRF) grant funded by the Korean government (MSIT) (2021R1C1C1013861)

## Author contributions

H.Y. participated in all data analyses, conclusions, and writing of the manuscript. B.P. performed all bioinformatics analyses and code writing. H.K. and H.J.J. contributed to the microbiome analysis. K.L.V. and J.A.M. oversaw all animal procedures. M.T.Y. performed blood analyses. M.A. contributed to metabolome analysis. M.B. and I.M. performed the cytokine assay. A.D.R. reviewed and edited the manuscript, I.B. supervised the project, and was involved in all aspects. I.B., A.D.R., and J.A.M. conceived and planned the project.

## Competing interests

The authors declare no competing interests.
