## [Peer Review File · Nature Communications]

REVIEWER COMMENTS

Reviewer #1 (Remarks to the Author):

Review for "Short-term periodic restricted feeding elicits metabolome-microbiome signatures with sex dimorphic persistence in primate intervention" by Dr. Beerman's laboratory.

Perhaps one of the hottest emerging science evoking broad interest in the scientific community and the general public is the topic of diet composition, calories, time-of-feeding, and their roles in influencing disease, healthspan and longevity regulation. Equally timely is the increased interest in dietary interventions and their potential interaction with pharmacological strategies to improve health and survival. In this context, the total calories consumed and the fasting/feeding times (when and for how long) emerge as key components of this equation for calories/health/survival across multiple species.

The manuscript presented here is a fantastic and well-executed set of studies focused on dissecting the role of a type of intermittent fasting, short-term cycles of restricted feeding, on mitigating some aging phenotypes of Rhesus Macaques. In this manuscript, the authors explore the effects, efficacy, and safety of Periodic Restricted Feeding in rhesus macaques (*Macaca mulatta*), by implementing a regimen of 4 days of low-calorie intake followed by 10 days of ad libitum feeding (4:10 diet) performed in repeating cycles over 12 weeks using middle-aged Rhesus macaques matched for age and sex. This PRF intervention resulted in a significant weight loss and fat percentage, even when the Ad libitum animals were fed for most of the study duration. The 4:10 diet, induced changes in lipid metabolism, and gut microbiome, showing a clear sex dimorphic response to the 4:10 diet intervention. Perhaps one of the most striking findings is the lasting effect of the intervention, suggesting a profound long-lasting remodeling of the microbiome-metabolome axes status.

This manuscript shows the potential for translating and implementing a novel dietary strategy based on intermittent fasting/Caloric restriction for the clinic using a proper pre-clinical model. These results are exciting and may help improve our understanding of how to implement this type of intervention into areas of clinical treatments where other strategies have miserably failed. Undoubtedly, more studies are needed to fully understand the long-term consequences and reduction of disease burden in this preclinical model using nonhuman primates, but still, this study provides a solid step forward. This work is of great interest to the readership of Nature Communications, the broader scientific community, and the general public.

Here are some specific comments that should be addressed/discussed in detail to improve the clarity and context of the findings:

1. To allow the scientific community to repeat and evaluate these results and dietary intervention, the authors must provide a detailed description/composition/implementation of the NHP FMD diet.

Were there any issues with the NHP adapting to the cycles of fasting/refeeding? I noticed that the animals undergoing the cycles did eat less than the total amounts of the Ad Lib group. What was the actual percentage of the total restriction over the 12 weeks?

2. Keeping on the diet theme, what the authors show here are the effects of the standard chow provided at the same feeding level for the same period of time. Are the AL animals “true” AL or, like in their CR studies fed two meals a day? Could the PRF then be interfering with their normal circadian rhythms? It would be important to discuss the potential impact on their circadian rhythmicity during and post intervention.

3. What are the levels of circulating β -hydroxybutyrate in blood across the different days of PRF?

4. Were there any T-cell composition/function, or cytokines measurements in the PRF studies? It would be important to report if the changes observed in these animals are comparable to those in mice and humans undergoing similar PRF/FMD interventions. Were there any functional tests done with them?

5. While properly acknowledged as observational. There is a bit of emphasis on the ability of PRF to alter body weight. How is this maintained in the long run, even after ceasing the dietary intervention? What could be driving this?

Reviewer #2 (Remarks to the Author):

Yanai et al. report in this manuscript the effects of priodic restriced feeding vs. adlib feeding on the monkey gut microbiome and the metabolome. While I find the overall study design interesting, the effects of caloric restriction (be it periodic or permanent) on the microbiome are already well described even in humans. To me the only advantage here is that food intake is highly controlled in monkeys which is hard if not impossible to do in humans. Apart form that, causal mechanistic studies how the observed changes may controbute to health are lacking. The authors conclude that the intervention appears safe and that now such interventions could also be done in humans - but I believe this study is not needed for such an intervention to be done in humans since it would be expected to be safe enough. Additionally, the 3 year follow-up data would have been interesting if no further intervention studies were done but this seems not to be the case and thus I fear these data are not interpretable.

Apart from this general evaluation my specific comments are listed below:

- Suppl. Fig. 6 does not show the schedule – page 12
- It is stated that the groups were matched for body weight. In Table 1 it seems as if the PRF group was approx.. 13% heavier. Was this difference statistically significant. If so the data would have to be corrected for this difference at baseline. How different were the animals in terms of lean mass and fat mass at baseline?
- Please avoid the word “striking” – page 4
- Fig. 1b right panel is interesting but if there are no data on what intervention studies were done with the animals during this time, the panel should be removed since no conclusion can be drawn from it. The same for Suppl. Fig. 4
- Fig. 2a: Day 14 is not indicated
- It is interesting that alpha diversity remains increased at C6d8 (Fig. 3 c) – I would expect a reduction in alpha div. upon refeeding. The authors should speculate what might be the reason for this finding.
- Since each group (AL and PRF) were housed in 3 groups, the authors should investigate if the microbiomes in the monkeys housed together were more similar than the others. If so, which I would expect, then this adds additional variation to these data and represents a limitation. The authors should show these data.
- Page 8: “Taken together, these results suggest that limiting calorie consumption, without altering the composition, leads to swift alterations in the microbiome that appear to be additive and are sustained beyond the conclusion of the dietary intervention.” Since there was no long-term follow up on the microbiome, this conclusion cannot be made on sustainability of the changes cannot be made.
- Suppl. Fig. 9: To me there is no convincing sex-specific increase.
- A section on the limitations of this study is missing

We would like to thank the Editor and Reviewers for their insight and comments to improve our manuscript evaluating the effects of short term PRF on non-human primates. We attempted to incorporate the suggestions and address the concerns of reviewers (as noted in the point-by-point response below).

REVIEWER COMMENTS

Reviewer #1 (Remarks to the Author):

Review for "Short-term periodic restricted feeding elicits metabolome-microbiome signatures with sex dimorphic persistence in primate intervention" by Dr. Beerman's laboratory.

Perhaps one of the hottest emerging science evoking broad interest in the scientific community and the general public is the topic of diet composition, calories, time-of-feeding, and their roles in influencing disease, healthspan and longevity regulation. Equally timely is the increased interest in dietary interventions and their potential interaction with pharmacological strategies to improve health and survival. In this context, the total calories consumed and the fasting/feeding times (when and for how long) emerge as key components of this equation for calories/health/survival across multiple species.

The manuscript presented here is a fantastic and well-executed set of studies focused on dissecting the role of a type of intermittent fasting, short-term cycles of restricted feeding, on mitigating some aging phenotypes of Rhesus Macaques. In this manuscript, the authors explore the effects, efficacy, and safety of Periodic Restricted Feeding in rhesus macaques (*Macaca mulatta*), by implementing a regimen of 4 days of low-calorie intake followed by 10 days of ad libitum feeding (4:10 diet) performed in repeating cycles over 12 weeks using middle-aged Rhesus macaques matched for age and sex. This PRF intervention resulted in a significant weight loss and fat percentage, even when the Ad libitum animals were fed for most of the study duration. The 4:10 diet, induced changes in lipid metabolism, and gut microbiome, showing a clear sex dimorphic response to the 4:10 diet intervention. Perhaps one of the most striking findings is the lasting effect of the intervention, suggesting a profound long-lasting remodeling of the microbiome-metabolome axes status.

This manuscript shows the potential for translating and implementing a novel dietary strategy based on intermittent fasting/Caloric restriction for the clinic using a proper pre-clinical model. These results are exciting and may help improve our understanding of how to implement this type of intervention into areas of clinical treatments where other strategies have miserably failed. Undoubtedly, more studies are needed to fully understand the long-term consequences and reduction of disease burden in this preclinical model using nonhuman primates, but still, this study provides a solid step forward. This work is of great interest to the readership of Nature Communications, the broader scientific community, and the general public.

Thank you for the positive feedback!

Here are some specific comments that should be addressed/discussed in detail to improve the clarity and context of the findings:

1. To allow the scientific community to repeat and evaluate these results and dietary intervention, the authors must provide a detailed description/composition/implementation of the NHP FMD diet. Were there any issues with the NHP adapting to the cycles of fasting/refeeding? I noticed

that the animals undergoing the cycles did eat less than the total amounts of the Ad Lib group. What was the actual percentage of the total restriction over the 12 weeks?

We have significantly expanded the methods section explaining how the diet was performed, including the addition of Suppl. Data Table 3 that details the dietary composition in a way that allows for replication of the experiment. We did not observe any issues with the NHP adapting to the diet, in terms of abnormal behavior that would indicate stress. It is important, however, to note that we attempted to design the diet to minimize stress and potential adaptability issues. Specifically, we gradually reduced food allotment (50% on day 1 and then more restrictive for the following days). To reduce binge feeding (but not overeating) on days five and six, rations were evenly divided and offered in small allotments provided approximately every two – three hours throughout the day (this was still AL amounts as they were provided their baseline AL + 50% on those days so would not prevent overeating). All of this is now better explained in Methods (see page 13, line 18 – page 14, line 11)

Regarding the total CR that animals underwent throughout the study, we re-examined the data to evaluate the effective CR that each animal received by calculating the amount that was actually eaten compared to the amount eaten during the baseline period. Overall, PRF animals underwent 70-90% effective restriction. This is now included in Suppl. Fig. 1f. This point also led us to an interesting evaluation of the correlation between total effective CR and BW loss. Surprisingly, the BW loss was not significantly correlated to effective total CR, suggesting that the BW loss effect of the PRF regimen is a different paradigm than simply “CR”. This result is now included in Fig. 1e and discussed in the results (Page 5, Lines 3-5) and Discussion (see page 9, lines 25-30).

2. Keeping on the diet theme, what the authors show here are the effects of the standard chow provided at the same feeding level for the same period of time. Are the AL animals “true” AL or, like in their CR studies fed two meals a day? Could the PRF then be interfering with their normal circadian rhythms? It would be important to discuss the potential impact on their circadian rhythmicity during and post intervention.

Thank you for pointing out this omission as we overlooked discussing this point in the initial submission. The AL animals are fed twice daily and are therefore also restricted in the feeding time, similar to the PRF subjects. The AL feeding includes as much food as desired during the feeding window. We hope the revised methods section now better explains this detail (see page 13, line 18 – page 14, line 11).

As during PRF, the restricted animals received only one feeding a day (also now better explained in the Methods), this may indeed have an impact on the circadian rhythm as suggested by the works of Froy et al. (for examples see ¹⁻⁴) as the PRF were fed mid-day rather than in the morning. Unfortunately, we did not quantify the circadian rhythm with the appropriate methodologies so our observations on impact to the circadian rhythm are speculative. That said, the animal handlers did not note any overt behavioral changes (including sleep disruption) that would indicate a circadian shift. We appreciate the suggestion to include this in the discussion (see page 12, lines 10-13).

3. What are the levels of circulating β -hydroxybutyrate in blood across the different days of PRF?

We measured the BHB levels with the Metabolon platform (see attached figure). We agree that it is an important metabolite to measure as it is an indicator of a shift from glucose to fatty-acid utilization and the ability of the brain to use BHB. We unfortunately only measure BHB at baseline, peak diet (day 4) of cycles 3 and 6, and at cycle 6 day 14 (after 10 days of return to AL) and can only conclude at this resolution. We now include a short description in the Results of the importance of this observation (see page 6, lines 1-4).

Reviewer Figure 1. Beta-hydroxybutyrate levels. Shown are the scaled intensities at baseline, C3d4, C6d4, and C6d14. The line is a spline correlation with confidence of fit (shaded)

4. Were there any T-cell composition/function, or cytokines measurements in the PRF studies? It would be important to report if the changes observed in these animals are comparable to those in mice and humans undergoing similar PRF/FMD interventions. Were there any functional tests done with them?

We regrettably did not perform T-cell functional assays: we were hoping to perform immunization challenges but were unable to perform these due to the COVID restrictions at the time. However, we did have some frozen plasma saved and have now included a NHP cytokine assay in the manuscript (see Suppl. Fig. 10; Results section, page 8, lines 15-17; Discussion, page 11, lines 16-17).

We did, however, examine T-cell composition via flow cytometry. While in general we did not see a strong impact of PRF on the immune system using the relatively broadly encompassing antibody panel to capture multiple lineages by flow cytometry analysis (i.e., wide vs deep). As T-cells are of particular interest, we have now included an unsupervised analysis of the flow-cytometry data, looking for a potential impact of PRF using the Citrus platform which enables to look for changes without the limitation of using canonical flow cytometry gating strategies. The results show that PRF *did* have an impact on two subpopulations: on in a subset of CD4 T-cells (decrease) and another within p16-high neutrophils (increase). These results are now added to the manuscript and the corresponding Suppl. Figures have been significantly expanded (see Suppl. Fig. 9; Results section, page 8, lines 11-17; Discussion, page 11, lines 12-13).

The mouse study that employed the same diet paradigm⁵ did not assay blood immunophenotypes and there is no study with the same dietary restriction published in humans- that we are aware of. Therefore, we have no real point of comparison with this specific paradigm. However, as other dietary interventions have been reported to impact the immune system^{6,7} we did initially hypothesize we would also see an effect on the composition of the blood cells and were surprised by the stability of the NHP immune profile in the PRF subjects.

5. While properly acknowledged as observational. There is a bit of emphasis on the ability of PRF to alter body weight. How is this maintained in the long run, even after ceasing the dietary intervention? What could be driving this?

For us, this question is one of the most interesting ones that arose from this study, which is why we emphasized the observation. When designing the study, we anticipated a metabolic and microbiome driven “memory” as was observed in the murine study⁵, perhaps supplemented by an immune signature. However, all of the parameters measured in the study surprisingly returned close to baseline levels relatively quickly. There was an indication of a memory in the microbiome but not in a manner significant enough to explain the retained BW phenotype. As a long-term follow-up was not initially planned, the tests to delineate the drivers were not performed.

While in the females we observed that the metabolome was likely involved in regulating the regaining of BW in females (as seen in the overcorrection of the metabolite profile at the last measured timepoint), the males returned to baseline levels but maintained the BW differential. Thus, we hypothesize that the sex-specific BW loss maintenance, even after return to a normal AL diet, could be driven by epigenetic changes in the gut, perhaps driven by microbiome disruption; perturbations in the neural regulation of hunger/satiety; or alterations of the endocrinological system. The latter is especially interesting due to the sex-specificity of the BW maintenance observed in this study. This speculation is now added to the Discussion (see page 10, line 10 – page 11, line 2).

Reviewer #2 (Remarks to the Author):

Yanai et al. report in this manuscript the effects of periodic restricted feeding vs. adlib feeding on the monkey gut microbiome and the metabolome. While I find the overall study design interesting, the effects of caloric restriction (be it periodic or permanent) on the microbiome are already well described even in humans. To me the only advantage here is that food intake is highly controlled in monkeys which is hard if not impossible to do in humans. Apart from that, causal mechanistic studies how the observed changes may contribute to health are lacking. The authors conclude that the intervention appears safe and that now such interventions could also be done in humans - but I believe this study is not needed for such an intervention to be done in humans since it would be expected to be safe enough. Additionally, the 3 year follow-up data would have been interesting if no further intervention studies were done but this seems not to be the case and thus I fear these data are not interpretable.

Apart from this general evaluation my specific comments are listed below:

Thank you for providing some key points that we did not clarify well in our initial submission. Please find below our response to the presented concerns.

Regarding the comment of how this intervention may contribute to health, we would like to emphasize that the diet resulted in a significant loss of body weight, which we now show was not directly correlated with the decrease in calorie consumption. This parameter (body weight loss) on its own has been used as an important indicator / biomarker associated with improved health outcomes, especially in the context of interventions that seek to delay or ameliorate late-life ailments^{8,9}. We also observed that the diet we employed promoted fat loss over lean body mass, which is also an important predictor of health in later ages¹⁰⁻¹². As such, we believe that dietary interventions that are sustainable could have a strong impact on public health. To avoid overstating

our conclusions – we have toned down the language of our statements throughout the manuscript on overall health improvements.

We are also excited to present some additional data from the 3-year follow up- we apologize that we were not initially clear in our description and did not include relevant information about the types of studies that monkeys (that were included in additional studies) were enrolled in. We believe that the observation, while not planned for, is important and surprising, and have compiled a detailed list of all studies that the animals participated in following the PRF study. As seen in the table below the studies performed were not intervention studies (with the exception of perhaps the treadmill study in which 5 animals were experimental, 1 PRF Female, 1 AL Female, and 2 AL males, 1 PRF Male) and would have no impact on the body weight of the animals.

Subject	Experiment
MB1: AL,F	Flu Vax Exp
MB2: PRF,M	17AE2 R5 Exp, TM Con, Epigenetics R2
MB3: AL,M	17AE2 R5 Exp, Epigenetics R2
MB4: PRF,M	PET Study, Pre-Diabetic Treatment (Invokana)
MB5: PRF,M	TM Exp, Flu Vax Control
MB6: PRF,M	Covid BAL, CV Echos, Flu Vax Control
MB7: PRF,M	PET Study, Covid BAL, BM BXs, Muscle BXs, Pre-Diabetic Treatment (Invokana)
MB8: PRF,M	Epigenetics R2
MB9: PRF,F	CV Echos, TM Con, Muscle BX
MB10: AL,F	Covid BAL, TM Exp, Flu Vax Control
MB11: PRF,F	CV Echos, TM Exp, Flu Vax Exp
MB12: AL,M	Covid BAL, CV Echos, TM Exp, Flu Vax Con
MB13: AL,M	Euthanized
MB14: AL,M	None
MB15: PRF,F	Covid BAL, BM BXs
MB16: PRF,F	CV Echos, Muscle BX, Flu Vax Control
MB17: AL,F	Euthanized
MB18: AL,F	Euthanized

MB19: PRF,F	None
MB20: PRF,F	TM Con
MB21: AL,F	None
MB22: AL,M	Covid BAL, 17AE2 R5 Exp, CV Echos, TM Exp, Muscle BX
MB23: AL,M	Covid BAL, Flu Vax Exp

Reviewer Table 1. Detailed individual animal study participation following the PRF study. A general description of each study:

Flu Vax – 90-day study, all animals received a flu vaccine, some included a novel adjuvant (exp).
 CV Echos – cardiovascular echocardiograms performed, typically at single time point.
 17AE2 – 12-week study, daily estradiol treatment to determine lowest effective dose.
 TM – our treadmill exercise project, lasts ~3 months per animal.
 PET study – animals sent to Bethesda for PET scan during methylphenidate challenge.
 Covid BALs – BAL samples collected at single time point.
 BM Bxs – bone marrow biopsies collected at single time point.
 Muscle Bxs – vastus muscle biopsies collected at single time point.
 Pre-diabetic treatment project – tests various diabetes drugs in NHPs.

As seen in the table, five animals alone participated in a study that could have impacted their BW (MB5 and MB11 from the PRF group; MB10, MB12, and MB22 from the AL group; participated in a 3-month treadmill study). However, it is evident that the BW differential remained regardless of study type when plotted individually:

Reviewer Figure 2. Long-term body weight follow-up stratified by subsequent study type. Body weight (BW is presented as change from that measured in baseline. Each dot represents a single measurement on a single animal. The line is a spline regression with confidence of fit (shaded). AL are shown in black and PRF in red. Data is stratified by animals that have not participated in any subsequent study (None), those that participated in non-intervention studies (No intervention), and those that participated in an treadmill study.

• Suppl. Fig. 6 does not show the schedule – page 12

Our apologies for this oversight- please now see Suppl. Fig. 11.

• It is stated that the groups were matched for body weight. In Table 1 it seems as if the PRF group was approx.. 13% heavier. Was this difference statistically significant. If so the data would

have to be corrected for this difference at baseline. How different were the animals in terms of lean mass and fat mass at baseline?

In our initial design we attempted to best match in the control and experimental groups “sex-, age-, body weight-, and fasting blood glucose assessed at baseline” (pg 13, line 13-14). We did however still have a mean difference in baseline BW between PRF and AL groups of 11.2%, as noted by the reviewer. As this can indeed have an impact on the effect of the diet, we re-examined the baseline data again, and shown in the figure below, the baseline BW difference was not significantly different. It is also important to note that the effects of the diet seemed to occur independently of starting BW, further supporting that the baseline difference was not critical (Suppl. Fig. 1b).

The question about body composition at baseline is important and we now have compared all the DEXA variables measured and as seen in the figure included here – the body composition between the groups is not different.

Reviewer Figure 3. Baseline weight and body composition. Left panel shows the individual baseline body weight (BW). Each dot is a single animal and the box indicates the mean and quartiles. Right panel shows the fat% as quartile box plots per depicted region.

- Please avoid the word “striking” – page 4

Done! We have changed “striking” to “notable”.

- Fig. 1b right panel is interesting but if there are no data on what intervention studies were done with the animals during this time, the panel should be removed since no conclusion can be drawn from it. The same for Suppl. Fig. 4

We agree that this data should have been shared from the onset and it is now included, please see above for details.

- Fig. 2a: Day 14 is not indicated

Corrected in the figure, thank you.

- It is interesting that alpha diversity remains increased at C6d8 (Fig. 3 c) – I would expect a reduction in alpha div. upon refeeding. The authors should speculate what might be the reason for this finding.

We too were surprised by this maintenance of diversity after refeeding (especially at the last time point, as it is divergent from the cycle 3 refeeding). We have two general hypotheses that are now described in Discussion (page 12, lines 6-13). The first is that the altered microbiome during peak diet promoted an alteration to the intestinal epigenetic landscape¹³ that in turn could have influenced the gut environment. This change was probably cycle dependent as it was only observed in the latest cycle. Due to the fact that we observed a sex-specific long-term maintenance of body weight, and the potential impact of sex on the microbiome and diet effects¹⁴, it is especially tantalizing to hypothesize an impact of the diet on the endocrinological system that feeds back to microbial diversity regulation. However, it is important to note that we did not observe a strong sex specific effect of the diet on the microbiome; but that could be because the number of animals after partitioning to sex is relatively low for robust statistical power.

- Since each group (AL and PRF) were housed in 3 groups, the authors should investigate if the microbiomes in the monkeys housed together were more similar than the others. If so, which I would expect, then this adds additional variation to these data and represents a limitation. The authors should show these data.

We apologize for not being more clear in describing the animal housing. The NHP were divided into batches (now detailed more clearly in Suppl. Fig. 10). It is important to note that these are not housing batches, but timing batches intended to render the work-load of animal handling feasible. All animals in the study were housed individually to allow for full regulation of feeding amounts. In addition, while samples were collected at different times, all the sequencing procedures were done simultaneously. However, we acknowledge that the experimental batches could have an impact and have now performed a batch-effect analysis. As seen in the newly added Suppl. Fig. 5 and attached here, we did not observe any significant batch effect on the microbiome profile, either at baseline or throughout the study.

Supplementary Figure 5. Batch effect test in the gut microbiome. a PCoA plots of all animals at baseline colored by batch as in b. **b** PCoA of all samples colored by batch. **c** beta-diversity test between each batch and all other batches.

• Page 8: “Taken together, these results suggest that limiting calorie consumption, without altering the composition, leads to swift alterations in the microbiome that appear to be additive and are sustained beyond the conclusion of the dietary intervention.” Since there was no long-

term follow up on the microbiome, this conclusion cannot be made on sustainability of the changes cannot be made.

We acknowledge our statement here was unclear – what we meant to state is that we saw indications of an altered microbiome at the end of the study (cycle 6, day 8) which was after the last restricted eating period. We should have been more precise. We edited the statement in Conclusion section to better reflect our interpretation of the data (page 9, line 3)

- Suppl. Fig. 9: To me there is no convincing sex-specific increase.

We were perhaps a little too excited about this particular compound and upon re-examination we agree that this result is not distinct enough – the figure has been removed and the Result section has been updated accordingly.

- A section on the limitations of this study is missing
Done!

Thank you again to all for the helpful comments and suggestions.

Additional References for Reviewer concerns

- 1 Froy, O. Circadian rhythms, nutrition and implications for longevity in urban environments. *Proc Nutr Soc* **77**, 216-222 (2018). <https://doi.org:10.1017/S0029665117003962>
- 2 Froy, O. The relationship between nutrition and circadian rhythms in mammals. *Front Neuroendocrinol* **28**, 61-71 (2007). <https://doi.org:10.1016/j.yfrne.2007.03.001>
- 3 Froy, O. & Miskin, R. The interrelations among feeding, circadian rhythms and ageing. *Prog Neurobiol* **82**, 142-150 (2007). <https://doi.org:10.1016/j.pneurobio.2007.03.002>
- 4 Froy, O., Chapnik, N. & Miskin, R. Effect of intermittent fasting on circadian rhythms in mice depends on feeding time. *Mech Ageing Dev* **130**, 154-160 (2009). <https://doi.org:10.1016/j.mad.2008.10.006>
- 5 Diaz-Ruiz, A. *et al.* Diet composition influences the metabolic benefits of short cycles of very low caloric intake. *Nat Commun* **12**, 6463 (2021). <https://doi.org:10.1038/s41467-021-26654-5>
- 6 Okawa, T., Nagai, M. & Hase, K. Dietary Intervention Impacts Immune Cell Functions and Dynamics by Inducing Metabolic Rewiring. *Front Immunol* **11** (2021). <https://doi.org:ARTN> 623989
10.3389/fimmu.2020.623989
- 7 Procaccini, C. *et al.* Caloric restriction for the immunometabolic control of human health. *Cardiovascular research* (2023). <https://doi.org:10.1093/cvr/evad035>
- 8 Stenholm, S. *et al.* Body mass index as a predictor of healthy and disease-free life expectancy between ages 50 and 75: a multicohort study. *Int J Obes (Lond)* **41**, 769-775 (2017). <https://doi.org:10.1038/ijo.2017.29>
- 9 Fontana, L. & Hu, F. B. Optimal body weight for health and longevity: bridging basic, clinical, and population research. *Ageing Cell* **13**, 391-400 (2014). <https://doi.org:10.1111/accel.12207>
- 10 Chang, C. S. *et al.* Effects of age and gender on body composition indices as predictors of mortality in middle-aged and old people. *Sci Rep* **12**, 7912 (2022). <https://doi.org:10.1038/s41598-022-12048-0>
- 11 Lee, D. H. & Giovannucci, E. L. Body composition and mortality in the general population: A review of epidemiologic studies. *Exp Biol Med (Maywood)* **243**, 1275-1285 (2018). <https://doi.org:10.1177/1535370218818161>
- 12 Mikkola, T. M. *et al.* Body composition as a predictor of physical performance in older age: A ten-year follow-up of the Helsinki Birth Cohort Study. *Arch Gerontol Geriatr* **77**, 163-168 (2018). <https://doi.org:10.1016/j.archger.2018.05.009>
- 13 Woo, V. & Alenghat, T. Epigenetic regulation by gut microbiota. *Gut Microbes* **14**, 2022407 (2022). <https://doi.org:10.1080/19490976.2021.2022407>
- 14 Valeri, F. & Endres, K. How biological sex of the host shapes its gut microbiota. *Front Neuroendocrinol* **61**, 100912 (2021). <https://doi.org:10.1016/j.yfrne.2021.100912>

REVIEWERS' COMMENTS

Reviewer #1 (Remarks to the Author):

The authors have addressed my comments in full, the manuscript has improved substantially since the last review. I have no further recommendations for the authors

Reviewer #2 (Remarks to the Author):

I thank the authors for the comprehensive addressing of my remarks.

In this light the follow-up body weight data are very interesting and novel. One wonders why food intake reduction was not associated with weight loss. If the microbiome is involved, it could be associated with changes in nutrient absorption, which has been shown in humans to be regulated by the gut microbiome (Basolo et al. Nature Medicine 2020).

This and the aspect of energy expenditure may play a role in the prolonged weight development. I would be excited to see this followed-up in future studies and have no further remarks.

Response to reviewers:

We would like to thank the reviewers for a productive review process and their constructive feedback – we feel the final submission is substantially improved from the initial manuscript.

Specific responses:

Reviewer #2:

“In this light the follow-up body weight data are very interesting and novel. One wonders why food intake reduction was not associated with weight loss. If the microbiome is involved, it could be associated with changes in nutrient absorption, which has been shown in humans to be regulated by the gut microbiome (Basolo et al. Nature Medicine 2020).

This and the aspect of energy expenditure may play a role in the prolonged weight development.”

Response: We also find the result quite exciting and think it definitely warrants further investigation. The potential impact to microbiome-dependent gut absorption changes is interesting. We added this hypothesis to the discussion (see page 12, lines 26-269) and will certainly keep it in mind for the future.

Same is true for the point about energy expenditure (see page 9, line 30 – page 10, line 2).